# A Robust DNA Isolation Protocol from Filtered Commercial Olive Oil for PCR-Based Fingerprinting

**DOI:** 10.3390/foods8100462

**Published:** 2019-10-09

**Authors:** Luciana Piarulli, Michele Antonio Savoia, Francesca Taranto, Nunzio D’Agostino, Ruggiero Sardaro, Stefania Girone, Susanna Gadaleta, Vincenzo Fucili, Claudio De Giovanni, Cinzia Montemurro, Antonella Pasqualone, Valentina Fanelli

**Affiliations:** 1SINAGRI S.r.l.—Spin Off of the University of Bari Aldo Moro, 70126 Bari, Italy; luciana.piarulli@libero.it (L.P.); micsav87@gmail.com (M.A.S.);; 2CREA Research Centre for Cereal and industrial Crops (CREA-CI) S.S. 673, km 25.200, 71122 Foggia, Italy; 3Department of Agricultural Sciences, University of Naples Federico II, via Università 100, 80055 Portici, Napoli, Italy; nunzio.dagostino@unina.it; 4Department of Agricultural and Environmental Sciences, University of Bari Aldo Moro, 70126 Bari, Italy; ruggierosardaro@gmail.com (R.S.); vincenzo.fucilli@uniba.it (V.F.); 5Department of Soil, Plant and Food Sciences, University of Bari Aldo Moro, 70126 Bari, Italy; sanna14@hotmail.it (S.G.); claudio.degiovanni@uniba.it (C.D.G.); antonella.pasqualone@uniba.it (A.P.); valentina.fanelli@uniba.it (V.F.)

**Keywords:** DNA extraction protocol, traceability, authentication, genetic tagging, SSRs, SNPs

## Abstract

Extra virgin olive oil (EVOO) has elevated commercial value due to its health appeal, desirable characteristics and quantitatively limited production, and thus it has become an object of intentional adulteration. As EVOOs on the market might consist of a blend of olive varieties or sometimes even of a mixture of oils from different botanical species, an array of DNA-fingerprinting methods have been developed to check the varietal composition of the blend. Starting from a comparison between publicly available DNA extraction protocols, we set up a timely, low-cost, reproducible and effective DNA isolation protocol, which allows an adequate amount of DNA to be recovered even from commercial filtered EVOOs. Then, in order to verify the effectiveness of the DNA extraction protocol herein proposed, we applied PCR-based fingerprinting methods starting from the DNA extracted from three EVOO samples of unknown composition. In particular, genomic regions harboring nine simple sequence repeats (SSRs) and eight genotyping-by-sequencing-derived single nucleotide polymorphism (SNP) markers were amplified for authentication and traceability of the three EVOO samples. The whole investigation strategy herein described might favor producers in terms of higher revenues and consumers in terms of price transparency and food safety.

## 1. Introduction

Olive (*Olea europaea* L. subsp. *europaea* var. *europaea*) is a species that originates from regions surrounding the Mediterranean Sea [1] and characterizes the landscape of the entire Mediterranean area [2]. The olive tree is mainly cultivated for oil production. More than 80% of olive oil production comes from the European Union, with Spain being the largest olive oil producer, followed by Italy and Greece (marketing years 2017/2018) [3]. However, a substantial decrease in olive oil production was observed in Italy in 2018, and it is expected to further decrease in 2019 [4] because of adverse weather conditions and phytosanitary issues that have plagued Italy in the last few years [5,6].

Extra virgin olive oil (EVOO) has elevated commercial value due to its health appeal, desirable characteristics and quantitatively limited production. As a high-value product, a certification label added to a bottle of EVOO allows food fraud to be prevented and consumers to trace oil “from tree to table”. The European Commission Regulation 1151/2012 disciplines quality schemes for agricultural products and defines guidelines on the labeling of foodstuffs.

Italy is the first ranked country in terms of the number of protected designation of origin (PDO), protected geographical indication (PGI) and traditional specialty guaranteed (TSG) certifications. As for EVOOs, these include 42 PDO and 4 PGI [7]. Given the importance of quality labeling, all PDO/PGI products have to adhere to specific reference standards. In the case of EVOOs, this means an array of rules on the use of specific olive varieties, the area and practices of cultivation, and the entire production chain from harvest to bottling. Consequently, EVOO traceability is a hot topic since it prevents intentional adulteration accomplished through the deliberate addition of low-cost edible vegetable oils of uncertain origin. As EVOOs on the market might consist of a blend of olive varieties or fraudulently sometimes even of a mixture of oils from different botanical species, varietal identification in olive oils is crucial and has attracted a growing interest in the last decade.

Several approaches based on chemical analysis have been proposed for EVOO traceability [8,9,10,11,12], which are useful to discover some types of macroscopic adulteration, such as the addition of oil from other species. However, they present some limits due to the difficulty in identifying the olive varieties used for oil production. Instead, DNA-based methods are more accurate and reliable than chemical methods; moreover, they are not influenced by environmental conditions [13]. Among the genetic markers used for cultivar identification and varietal traceability in processed foods, simple sequence repeats (SSRs) and single nucleotide polymorphisms (SNPs) are the most suitable [14,15,16]. Indeed, the use of molecular markers for olive oil traceability is widely documented [17,18,19,20,21].

To have a reliable DNA extraction protocol from EVOOs is the key for a successful molecular fingerprinting. A critical aspect to take into consideration concerns the recovery of an adequate amount of DNA from filtrated oils. DNA is usually degraded and present in a very low amount when a food matrix is used for its extraction [22,23]. So far, diverse methods have been developed to obtain high quality DNA from cellular sediment as well as from the oily and the aqueous phases of filtered EVOOs [24]. All of them share three fundamental steps: (i) cellular lysis; (ii) DNA recovery and (iii) DNA purification. Although each method differs from the other in terms of quality and amount of recovered DNA, all lack information on the effectiveness of the protocol in industrial/legal applications. Moreover, the ideal DNA extraction method should be reproducible, simple, relatively cheap and capable of recovering stable and PCR inhibitor-free DNA from heavily processed plant oil, both filtered and unfiltered. Although different DNA-based methods have been applied for EVOO authentication and traceability, all of them suffer the difficulty of standardizing the DNA extraction protocol as well as the subsequent amplification and genotyping steps.

Within this motivating context, our work presents a case study aiming at set up a timely, low-cost, reproducible and effective DNA extraction protocol from three filtered commercial EVOOs, which allows an adequate amount of DNA to be recovered. Different authors showed the effectiveness of extraction buffer containing CTAB and/or hexane in DNA isolation from different vegetable oils [23,24,25,26,27,28]. Based on these results, we decided to compare the n-hexane protocol by Consolandi et al. [29] with three CTAB-based methods by Muzzalupo and Perri [30], Busconi et al. [31], and Spadoni et al. [32]. CTAB residual could affect the downstream enzymatic reactions; therefore the detection of an effective and reproducible protocol for DNA extraction and amplification is of crucial importance. Furthermore, as reported by Schrader et al., 2012 [33], different salts (e.g., sodium chloride or potassium chloride), detergents or organic molecules (ethylene-diamine-tetra-acetic acid, sarkosyl, ethanol, isopropyl alcohol or phenol) necessary for efficient cell lysis or for the isolation of pure nucleic acids, might also cause PCR inhibition at certain concentrations. A secondary but no less important purpose was to verify the effectiveness of the DNA extraction protocol herein proposed and to assess PCR-based fingerprinting methods for olive oil authentication and traceability. To this end, we analyzed the DNA polymorphisms of nine SSRs and the presence/absence of eight SNP markers for genetic tagging of Italian EVOOs.

## 2. Materials and Methods

### 2.1. Oil Samples and DNA Extraction

Analyses were performed on three EVOO samples (hereinafter referred to as OL1, OL2, and OL3), provided by an Italian mill, which underwent two filtration steps in order to extend their shelf-life. The first step was carried out with two filters made of cellulose and diatomaceous flours; the second filtration step was performed using a polishing paper. The only available information on those samples was that they were bottled in Italy. Three bottles of one liter each were collected and stored at 4 °C before the analysis.

We tested four protocols (hereinafter referred to as p1, p2, p3, and p4) for DNA isolation:p1 Spadoni et al., 2019 [32];p2 Muzzalupo et al., 2002 [30];p3 Busconi et al., 2003 [31];p4 Consolandi et al., 2008 [29].

After evaluating those protocols, we modified p4 as follows:One mL per sample was vigorously mixed for 5 min and transferred into a 2 mL microcentrifuge tube, then it was shaken with 500 µL of hexane for 5 min and centrifuged (15 min, 4 °C, 12 krpm).The oily supernatant was transferred to a fresh 2 mL tube and the pellet was mixed with 400 µL of lysis buffer (50 mM Tris pH 8.5, 1 mM EDTA, 0.5% *v*/*v* Tween 20).Samples were incubated at 48 °C for 1 h and then centrifuged (15 min, 4 °C, 12 krpm).The two aqueous phases from the oil/buffer mix and the pellet were transferred into separate 1.5 mL microcentrifuge tubes, being careful to avoid contact with the interface between oil and buffer.The pellet derived from treatment of the aqueous phase at point 4 was discarded.The three phases were transferred into separate 1.5 mL microcentrifuge tubes and the samples were gently mixed with 500 µL of cold isopropanol, held 30 min at −80 °C, and centrifuged again (30 min, 4 °C, 12 krpm).The pellet was washed by adding 500 µL of absolute ethanol, held 30 min at −80 °C and centrifuged (30 min, 4 °C, 14 krpm).The pellet was washed once more by adding 500 µL of 70% ethanol and centrifuged (30 min, 4 °C, 14 krpm).After removing the supernatant, the pellet was dried in vacuum pump for 30 min and finally resuspended in 50 µL of 1× TE buffer pH 8 (10 mM Tris, 1 mM EDTA).

### 2.2. SSRs and SNPs Assay

A set of nine SSR primer pairs [34,35,36] was used to assess the genetic fingerprinting of the three oil samples. Of these, seven (i.e., DCA03, DCA05, DCA09, DCA18, GAPU71B, GAPU101, EMO90) were selected among those available in literature [37] due to their high reproducibility, power of discrimination and number of amplified loci/alleles. The remaining two (i.e., DCA04 and DCA15) were included as they have been proven to be suitable for PCR-based fingerprinting methods in olive [38,39]. PCR was carried out following the procedure by di Rienzo et al. [40]. Sequences and annealing temperatures of each primer are reported in Appendix A.

Amplification products were separated by capillary electrophoresis on the ABI PRISM 3100 Avant Genetic Analyzer sequencer (Applied Biosystems, Foster City, CA, USA) using a mix containing 2 µL PCR reaction, 13,5 µL Hi-Di™ Formamide and 0.5 µL GeneScan™ 600 LIZ™ Size Standard (Applied Biosystem, Foster City, CA, USA). Electropherograms were analyzed by the GeneMapper Software v3.7 (Applied Biosystems, Foster City, CA, USA).

A large panel of 470 Mediterranean olive accessions (from the private database developed by the Department of Soil, Plant and Food Sciences, University of Bari Aldo Moro) including Italian (396), Tunisian (25), Syrian (23), and Algerian (26) accessions, was used to assess genetic similarity among olive accessions and the three commercial EVOOs.

Genetic similarity between the three EVOO samples and the 470 olive accessions was calculated through principal coordinates analysis (PCoA) using the GenAlEx software version 6.5 (http://biology-assets.anu.edu.au/GenAlEx/Download.html). Based on PcoA, accessions with SSR-based genetic profiles more similar to the three samples under study were selected for the construction of a tree through the un-weighted neighbor-joining method, running 1000 bootstrap replicates, using DARWIN v 6.0.010 (http://darwin.cirad.fr) [41]. The resulting tree was visualized using FigTree 2016-10-04-v1.4.3 (http://tree.bio.ed.ac.uk/software/figtree/).

A total of six Italian olive cultivars (Frantoio, Taggiasca, Pendolino, Crastu, Leccino, and Ogliarola barese), representative of the main PDO and PGI certifications (http://www.agraria.org/prodottitipici/oliodioliva.htm), were selected for SNP analysis. We took in consideration the SNP profile from a pre-existing catalogue generated by genotyping-by-sequencing [42]. We selected eight SNPs that satisfied two requirements: (i) each SNP should be polymorphic among the six cultivars; and (ii) the allelic profiles detected using eight SNPs should be specific for each cultivar. Eight primer pairs were designed in SNP flanking sequences.

PCR was performed in a final volume of 20 µL containing 50 ng of DNA, 1X Phusion HF Buffer, 0.2 mM dNTPs Mix, 0.4 µM primer mix and 0.5 units of Phusion high-fidelity DNA polymerase (Thermo Fisher Scientific, Waltham, MA, USA). The cycling program was: 2 min of initial denaturation at 98 °C, 35 cycles of denaturation at 98 °C for 10 sec, annealing for 20 sec, extension at 72 °C for 30 sec and final extension at 72 °C for 5 min. Annealing temperatures ranged from 64 to 68 °C. PCR products were analyzed by 2% agarose gel electrophoresis to verify the specificity of the amplification reaction. Each amplicon was then purified through ethanol precipitation and quantified using the NanoDropTM ND2000C (Thermo Fisher Scientific, Waltham, MA, USA).

PCR products obtained from DNA extracted from oils required an additional step of purification, which was performed using the Agencourt AMPure XP - PCR Purification Kit (Beckman Coulter, USA). Sequencing reaction was prepared using the BigDye™ Terminator v3.1 Cycle Sequencing Kit following the manufacturer’s instructions and the sequencing was performed by the ABI PRISM 3100 Avant Genetic Analyzer (Applied Biosystems, Foster City, CA, USA).

Finally, raw data were analyzed by the Sequencing Analysis Software v6.0 and the Sequence Scanner Software v2.0. 

## 3. Results

### 3.1. Set Up of An Effective Protocol for DNA Extraction from Filtered Oils and EVOO Traceability

As a first step, the four protocols (i.e., p1–p4) were compared each other to assess their effectiveness in extracting DNA from the three filtered EVOO samples. This was done to identify the most suitable one to be modified if necessary (Table 1). Although p1 was developed to extract DNA from olive leaves, we have also chosen to assess its effectiveness because of its rapidity and flexibility.

All protocols share an initial step of sample incubation at 60–65 °C with a standard extraction buffer consisting of Tris-HCl and EDTA. All protocols, except p4, are based on the use of CTAB, while p1 includes the addition of the anionic detergent sodium dodecyl sulfate (Table 1). Afterwards, organic solvents such as hexane, dichloromethane, octanol, and phenol were used for the dissolution of the plasma membrane. DNA precipitation was obtained with ethanol or isopropanol.

Each protocol was tested following authors’ indications. After extraction, DNA was checked for quantity and quality. In all cases, DNA was highly degraded and its amount ranged from 0.5 ng/μL (p1) to 100 ng/µL (p4).

P1 was discarded because of the low amount of recovered DNA (<2 ng/μL). P2 and p3 were both based on the use of CTAB. Although both protocols start from a volume of oil higher than 10 mL, the amount of recovered DNA was lower than 10 ng/μL. Moreover, the use of CTAB, which is an inhibitor of enzymatic reactions, requires several washing steps to remove its traces. By contrast, the extraction solvent used in p4 is the hexane and this protocol resulted in a fair amount of DNA, ranging from 20 to 100 ng/µL, though starting from 2 mL of EVOO. The only drawback is that p4 was the most time-consuming. Therefore, we decided to improve p4 in order to (i) reduce the time required for the DNA extraction process; (ii) decrease the starting amount of oil; and (iii) limit the number of reagents and equipment needed. All these improvements are crucial for industrial and legal application of the method.

The protocol herein proposed (labeled as p5 in Table 1) was obtained through the optimization of p4 with some key modifications. Firstly, a reduction of the starting material (to 1 mL) was applied; this change allowed the starting centrifugation step included in p4 to be eliminated and replaced by vigorous stirring for a few minutes in a ThermoMixer. Furthermore, the sample digestion step with Proteinase K was ruled out, thus reducing to 1 h the incubation time of sample after the addition of the lysis buffer. Finally, two steps of DNA precipitation were performed. The first one was carried out in isopropanol with incubation at −80 °C for 30 min instead of −20 °C overnight. The second step was performed in absolute ethanol with incubation at −80 °C for 30 min. Nucleic acid quality evaluation was performed by 1% of agarose gel (Appendix A) and showed a high degradation of genomic DNA. The quantity values, checked through Nanodrop, ranged between 53.8 ng/μL and 442.7 ng/μL for DNA extracted from oily and aqueous phases and between 0.4 ng/μL and 9.5 ng/μL for DNA extracted from the pellet phase. 

### 3.2. EVOO Genotyping and Traceability

To test the suitability and effectiveness of the protocol herein proposed and set up a reproducible genetic tagging and authentication method, genomic regions harboring SSR and SNP markers were amplified starting from the DNA extracted from the three oil samples. For each sample, three independent DNA extraction replicates were performed, in order to assess the repeatability of the technique. 

First of all, in order to evaluate the suitability of the extracted DNA to PCR amplification, we performed loci specific PCR amplifications using the microsatellite markers DCA03 and DCA18, which have proven to be highly reproducible based on our experience [43]. PCR products were obtained using DNA extracted from oily, aqueous and pellet phases (Appendix A), showing 100% repeatability (10 technical replicates). All samples provided a clear and distinguishable amplification pattern (Figure 1). 

For practical reasons, PCR runs for all SSR markers were performed on DNA extracted from aqueous phase only. The three EVOO samples showed monomorphic profiles for DCA04, DCA09, EMO90, and GAPU71B, while different allele combinations were recorded for the remaining markers in each sample (Figure 2a). The most informative markers were DCA03 and DCA18, for OL1 and OL3, respectively, as each of them discriminated among 10 different allele combinations. The total number of alleles was 22 for all oils, even the allelic variants were different. We compared the pattern of SSR allele sizes with a database of SSR-based allele frequencies of 470 records including Italian, Tunisian, Algerian, Syrian, and native Apulian varieties, developed at the Department of Soil, Plant and Food Sciences, University of Bari “Aldo Moro” (Figure 2b).

We included a set of 38 olive accessions collected in Apulia, which is the major oil-producing region in the Southern Italy [39]. Indeed, many common national varieties used in the EVOO production (i.e., Coratina, Cima di Bitonto, Ogliarola barese, etc.) derived from Apulia. 

To explore genetic similarities among the three EVOO samples and the 470 olive accessions, principal coordinates analysis (PCoA) was performed (Figure 3).

The three samples resulted to be different but very close each other, suggesting the use of common and related varieties in EVOO production. Moreover, all the samples were placed in a group that includes the main Italian olive varieties. A subset of 147 accessions that lie in the fourth quadrant of the PCoA plot together with the three EVOO samples was filtered out and used to construct a tree through the un-weighted neighboring joining method (Figure 4). The dendrogram disclosed that the three EVOOs were grouped in the same cluster; however, OL1 and OL3 are more similar to each other. Finally, the clade in which the three EVOO samples fell includes some of the most widespread Italian varieties used for oil production, such as Frantoio, Cima di Bitonto and other many olive accessions originated from Apulia.

Eight SNPs, which discriminate against six varieties, namely Frantoio, Taggiasca, Pendolino, Crastu, Leccino, and Ogliarola barese, were selected and the DNA regions harboring those SNPs subjected to PCR amplification and Sanger sequencing. PCR experiments were successful in all the samples (Table 2). 

The presence of non-nucleic acid contaminants and inhibitors in the amplified DNA involved an additional purification step, following which sequencing reactions definitely improved as shown by the severe reduction of the background noise (Appendix A).

All samples were characterized by the same alleles at seven SNP sites (from SNP #1 to #7) with the exception of OL3, which showed a different allele for SNP #7 (Figure 5). As for SNP #8, differences in the allelic profile were found across all samples (Figure 6). 

## 4. Discussion

When olive drupes are ground, the cellular compartmentalization is destroyed and everything is mixed in the aqueous solution (i.e., waste water) where most of the DNA and proteins are. Olive oil is the water-insoluble fraction mainly lacking in DNA and proteins, but enriched in organic compounds that interfere with DNA polymerase reactions.

Genetic traceability of EVOOs requires DNA to be extracted from commercial olive oils, which are all subjected to a filtration process in order to satisfy consumers’ demand (i.e., clear oil).

To identify an effective and robust DNA extraction protocol from filtered commercial oils, four protocols were tested and compared each other (Table 1). However, none of these proved to be highly reproducible as the extraction of DNA from a complex matrix to be subjected to PCR is not a trivial task [15,44,45]. Following the comparison, only p4 turned out to be the most suitable in terms of DNA yield, but it did not guarantee repeatable results with regards to DNA amplification. Therefore, we improved p4 with key modifications in order to enhance the quality and quantity of the extracted DNA while reducing the starting material, reagents, costs and time of extraction. The protocol herein proposed (labeled as p5 in Table 1) was applied on three EVOO samples of unknown composition. Nanodrop measurements were satisfactory for the DNA extracted from the oil and aqueous phases with values ranging from 442.7 ng/μL to 53.8 ng/μL. By contrast, DNA concentration from the pellet phase ranged from 9.5 ng/μL to 0.4 ng/μL. Taking into account the nature of the starting material, we have obtained appreciable quantity of DNA from all three extraction phases.

The effectiveness of p5 was assessed by applying PCR-based fingerprinting methods for EVOO authentication and traceability. DNA amplifications of nine SSR loci were successful for all the three EVOO samples.

The presence of microsatellite multi-allelic combinations (Figure 2) coupled with results from PCoA and genetic similarity analysis (Figure 3 and Figure 4), both describing the relationships between the three EVOO samples and a large panel of accessions cultivated in the Mediterranean area, allowed us to assert that most likely the three EVOO samples are a blend of Italian varieties.

If SNP markers are used for EVOO authentication and traceability, we suggest using an additional purification step of PCR products. Indeed, phenolic compounds and residual polysaccharides could inhibit the enzymatic reactions providing irregular PCR amplifications [46]. This inhibition is negligible in routine PCRs, such as those for SSR amplification, but it is exacerbated in highly sensitive reactions, such as Sanger sequencing. Then, we applied an additional PCR purification step based on the use of magnetic beads after ethanol precipitation, which allowed us to gain a significant improvement in sequencing results (Appendix A). Previous works performed magnetic bead-based purification before PCR [47,48]; however, this procedure leads to a massive loss of DNA. As the amount of DNA extracted from filtrated oil was generally low, we decided to carry out this step after PCR amplification. 

Advances in technologies for DNA profiling have had a huge impact in combating fraud and food adulteration as well as in improving authentication and traceability of food products. The screening of multiple markers with next generation sequencing (NGS) technology has been widely applied to the olive sector, and in particular the high-resolution DNA melting technology (HRM) has been successfully used for olive oil authentication [49,50]. HRM analysis using gene-based SNP markers represents an important tool for cultivar identification. However, gDNA extraction from olive oil matrix is still a critical step, which needs improvement [51]. For that reason, we chose to amplify the DNA of the three EVOOs with SNP-based markers and then perform sequencing by Sanger method. SNP-based molecular profiles revealed high similarity among the three EVOO samples, thus indicating that they shared some varieties, as observed before. By comparing these profiles with those obtained for six Italian olive cultivars, we observed that OL1 and OL2 were very similar to Leccino, while OL3 is more similar to Frantoio and Taggiasca (Table 2).

Despite no information about varietal composition of the three EVOO samples under investigation was available, we were able to assert that most likely the three samples are blends of Italian varieties and generate a list of possible varieties present in each blend. 

The EU commission regulation 182/2009 governs the whole production process of EVOOs. The origin of olives and/or the addition of oils coming from different countries must be indicated on the label. Molecular traceability is an important tool to investigate the varietal composition of olive oils, in order to prevent fraud due to the use of oils coming from extra-EU countries not explicitly indicated on the label.

Contrary to what happens for the wine market, in which there are few international varieties (i.e., Cabernet, and Merlot) that are cultivated widespread in the world, for the oil market there are cultivars strongly adapted and cultivated in each territory (i.e., Arbequinia and Arbosana for Spain, Chemlali for Tunisia, Coratina and Frantoio for Italy). In this context, the identification of varietal composition could be useful to better define the origin of the oil.

Nevertheless, the comprehensive identification of all the varieties present in a blend is extremely tricky. The identification of singleton SNPs and/or haplotype blocks via NGS will allow the rapid detection of cultivars present in a blend in the near future.

## 5. Conclusions

We developed a speedy and highly reproducible protocol able to recover whole genomic DNA from filtrated EVOOs. Then, we applied PCR-based methods for genetic tagging of filtered EVOOs of unknown origin in order to verify the effectiveness of the proposed DNA extraction protocol. The whole investigation strategy may be suitable for industrial/legal applications, as olive oil frauds are more and more rampant. Deliberate or unintentional adulteration of olive oil could seriously damage producers/providers in term of earnings and reputation and could definitely break the producer–consumer relationship. On the other hand, consumers are increasingly demanding as their preferences are moving towards the highest quality products. There is a vast range of olive oil on the market and consumers would like to have awareness of the entire olive oil supply chain, “from tree to table”. DNA-based methods for olive oil authentication and traceability are beginning to be routinely applied, and although they are still challenging due to the complexity of fraudulent practices, they can reassure consumers in terms of price transparency and food safety.

## Figures and Tables

**Figure 1 foods-08-00462-f001:**
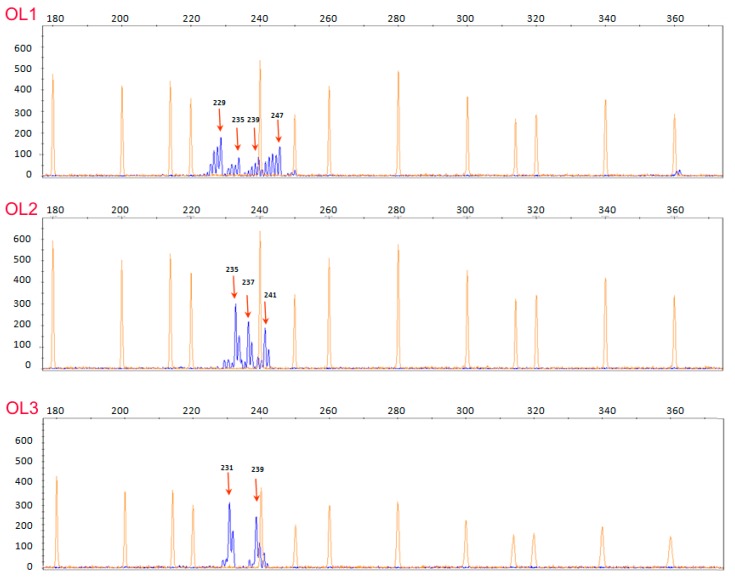
Capillary electropherograms showing the dinucleotide DCA03 amplification pattern for OL1, OL2 and OL3. Allele size (in base pairs) is indicated in correspondence of the main peak.

**Figure 2 foods-08-00462-f002:**
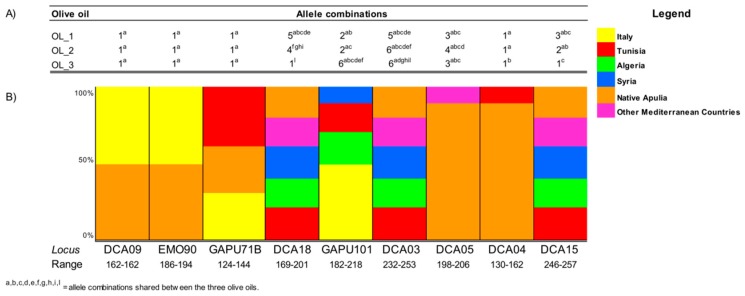
Allelic patterns detected in the three olive oils using nine simple sequence repeat (SSR) markers. (**A**) The number and composition of allelic combinations were reported for each extra virgin olive oil (EVOO) sample. (**B**) Bar plot representing the proportion of allelic combination for each of the nine SSR *loci*.

**Figure 3 foods-08-00462-f003:**
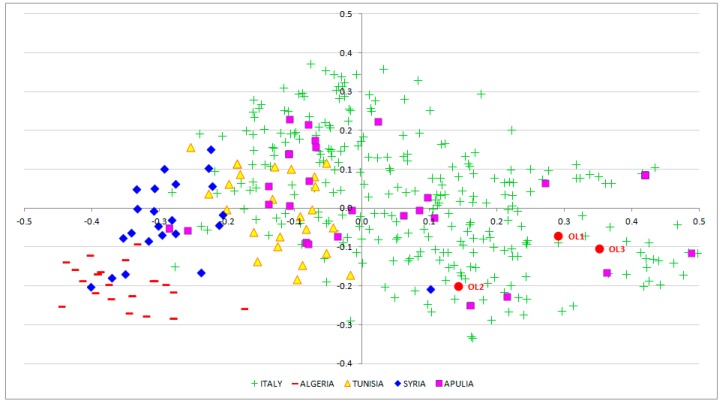
Principal coordinates analysis (PCoA) performed using the three EVOO samples and the panel of 470 Mediterranean olive accessions. OL1, OL2 and OL3 are marked in red.

**Figure 4 foods-08-00462-f004:**
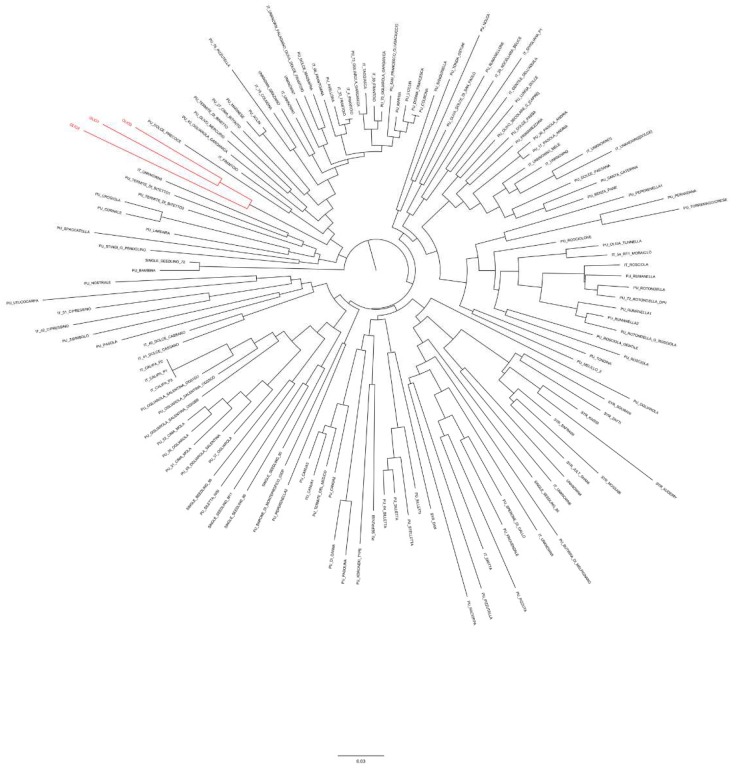
Genetic similarity between the three EVOO samples and the subset of 147 olive cultivars selected on the bases of PCoA results. OL1, OL2 and OL3 are marked in red.

**Figure 5 foods-08-00462-f005:**
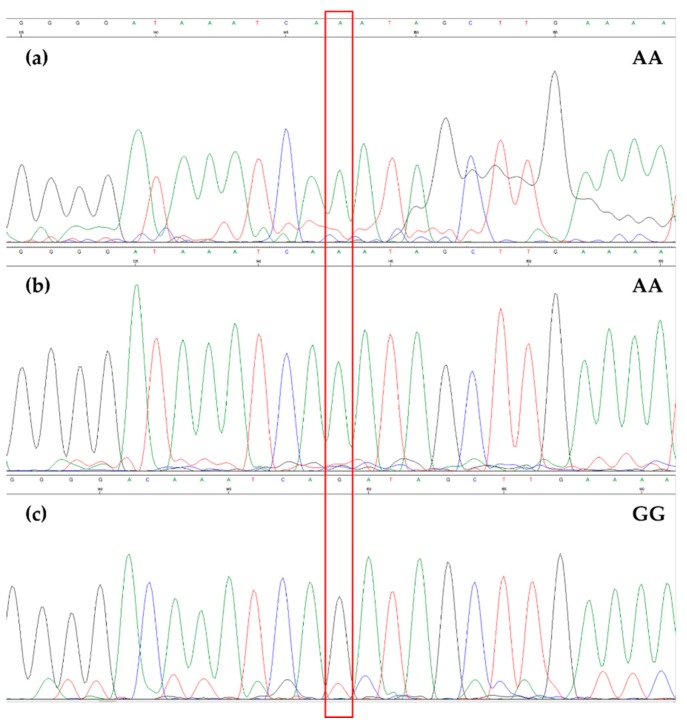
Electropherogram obtained from Sanger sequencing of the region containing SNP #7 in OL1 (**a**), OL2 (**b**), and OL3 (**c**). A red box highlights the SNP.

**Figure 6 foods-08-00462-f006:**
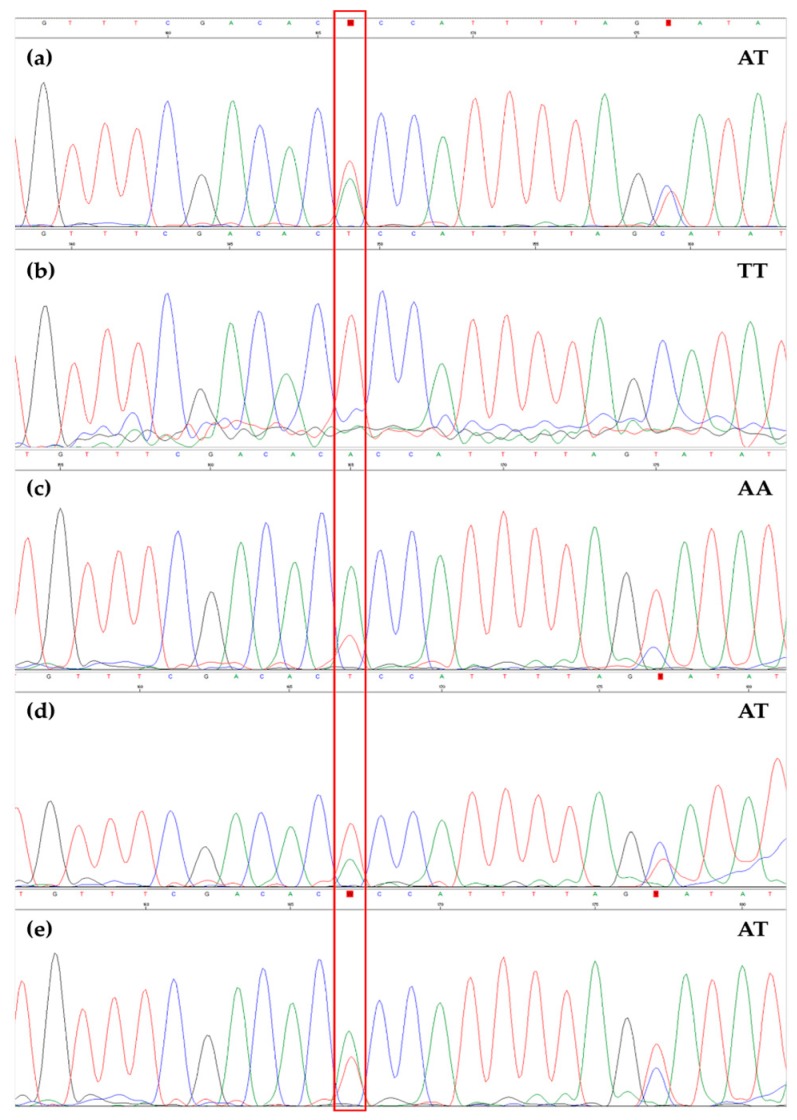
Electropherogram obtained from Sanger sequencing of the region containing SNP #8 in Ogliarola barese (**a**), Pendolino (**b**), OL1 (**c**), OL2 (**d**), and OL3 (**e**). A red box highlights the SNP.

**Table 1 foods-08-00462-t001:** Comparison among the DNA extraction methods used in this study. Extraction time, starting amount of plant tissue/oil (* as recommended by the authors), and reagents used are indicated.

	P1	P2	P3	P4	P5
	Spadoni et al., 2019 [32]	Muzzalupo et al., 2002 [30]	Busconi et al., 2003 [31]	Consolandi et al., 2008 [29]	Consolandi et al., 2008 modified [29]
Extraction time (hours)	~6 h	~30 h	~6 h	~30 h	~4 h
Sample/tissue and amount *	2 g leaves	10 mL not filtered, clear oil	50 mL not filtered, clear oil	2 mL not filtered, clear oil	1 mL filtered, clear oil
Starting centrifugation step	No	Yes	Yes	No	No
Main solvent	CTAB, phenol, chloroform	CTAB, dichloromethane, chloroform	CTAB, octanol, chloroform	Hexane, chloroform	Hexane, chloroform
Liquid nitrogen	No	No	Yes	No	No
Proteinase K	No	No	No	Yes	No
Pronase	No	Yes	No	No	No
SDS	Yes	No	No	No	No
RNAse treatment	No	Yes	Yes	No	No
β-mercaptoethanol	Yes	Yes	Yes	No	No
Extraction time (hours)	~6 h	~30 h	~6 h	~30 h	~4 h

**Table 2 foods-08-00462-t002:** List of single nucleotide polymorphisms (SNPs) validated in the selected varieties as well as in the three EVOO samples. n.a. = not available.

Variety	# SNP							
	1	2	3	4	5	6	7	8
Frantoio	CC	GG	TC	GG	AT	GG	GG	AT
Taggiasca	CC	GT	TC	GG	AT	GC	GG	AT
Pendolino	CC	GG	TT	AG	TT	GC	AA	AA
Crastu	TT	TT	TT	AG	TT	GG	AA	AA
Leccino	CC	GG	TC	GG	TT	GG	AA	AT
Ogliarola barese	CC	GT	n.a.	GG	AT	CC	GG	AT
OL1	CC	GT	TT	GG	TT	GG	AA	AT
OL2	CC	GT	TT	n.a.	TT	GG	AA	AT
OL3	CC	GT	TT	GG	TT	GG	GG	AT

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
