# Peer review of "A Robust DNA Isolation Protocol from Filtered Commercial Olive Oil for PCR-Based Fingerprinting"

_foods, 2019, doi:10.3390/foods8100462_

Round 1

Reviewer 1 Report

The manuscript entitled “A robust DNA isolation protocol from filtered 2 commercial olive oil for PCR-based fingerprinting” submitted to Foods MDPI, under the ID: foods-594343, is an interesting manuscript, well written and under a very important and actual thematic. However. There are some situations that are less clear, or that require reformulation since the statements/approaches may not be completely correct.

The abstract needs to be rewritten mainly because the authors are bit harsh when they state “ EVOOs on the market frequently consist of a blend of olive varieties or 18 even of a mixture of oils from different botanical species, an array of DNA-fingerprinting methods 19 have been developed so far for their authentication.” The EVOOs on the market are not frequently from a mixture of oils from different botanical species. I believe that this sentence is offensive for the olive oil producers.

Another point that requires a deeper review by the authors is the bibliography that is very self-concentrated. There are references that are unnecessary, since they already have reference that support such statements (example reference 21). More, there are some other important papers in the area that have not been referred to, inclusively that have already worked with commercial olive oils using PCR-based methods apart from reference 22 (Martins-Lopes et al. (2008). DNA markers for Portuguese olive oil fingerprinting. Journal of Agricultural and Food Chemistry, 56(24): 11786–11791. DOI: 10.1021/jf801146z and Gomes et al. (2018). Microsatellite High-Resolution Melting (SSR-HRM) to Track Olive Genotypes: From Field to Olive Oil. Journal of Food Science, 8: 2415-2423. Doi: 10.1111/1750-3841.14333).

In line 83 of the manuscript, when the authors refer that CTAB can affect enzymatic reactions, it is true but the authors should also include the reference to other chemicals that compromise the reaction success.

In line 100, in the material and method section, the p1 protocol doesn´t make sense to be included, since it a protocol driven for leaf samples. I recommend that it is taken out of this study. However, there are two important protocol that have not been tested, such as: Giménez, M.J., Pistón, F., Martín, A., Atienza, S.G. (2010). Application of real-time PCR on the development of molecular markers and to evaluate critical aspects for olive oil authentication. Food Chemistry 118 (2): 482–487. http://doi.10.1016/j.foodchem.2009.05.012 and Raieta, K., Muccillo, L., Colantuoni, V. (2015). A novel reliable method of DNA extraction from olive oil suitable for molecular traceability. Food Chemistry, 172, 596–602. http://doi.org/10.1016/j.foodchem.2014.09.101), to which the authors have not compared or cited their work.

In table S1 it would be important to add the allele range size of each marker. This is important to discuss the SSR choice when dealing with DNA from processed samples, such as the ones analyzed in this study.

The choice to use Sanger sequencing in this type of approach is not recommended, the sequence analysis should be pursued using a technology that would reduce substantially the sample handling. The protocol used for SNP detection is time consuming and requires multiple steps, demonstrating it is not suitable to be applied in olive oil authenticity, since it would be more subjected mislabeling situations. More, there is a lack of precision in the varietal identification, which is due to the lack of several conditions required: (i) reference material (that needs to be from certified olive groves, so it can be used); (ii) the database would have to have information from other regions besides the described (Spanish, Greek, French, etc); (iii) there is a lack of statistical procedures or algorithms to confirm some of the statements referred in the text. The authors should be more careful with the identifications they claim to have.

In line 181 the authors refer that the DNA is highly degraded. How did they check this?

The amount of DNA recovered by the authors in each phase is not clear and more they should have extracted DNA from the same sample several times (at least 3) to have some kind of knowledge of the reproducibility of the method within a sample.

The selection of the SSRs in line 210 are not well justified.

Figure 1 a) does not have a clear pattern as stated in the text (line 215).

All the text referring Figure 2 cannot be validated since there is no Figure 2 in the manuscript.

Figure 4 is not really informative, since no probabilistic issue can be assessed once all the olive oil samples are blind, and therefore we cannot guarantee that they are not blends, we cannot guarantee that they only have Italian varieties, and we cannot guarantee that all the alleles have been correctly attributed. I believe that the authors have tried to demonstrate something that is not reliable, that is the composition of the olive oils assessed, and that is not a key question in this paper. The paragraph stating in line 314 is there abusive.

The reactions used to detect the SNPs were cleaned after the PCR. In this case how can we guarantee that the inhibitors of extraction protocol did not influence the overall result?

Reviewer 2 Report

This is an interesting and well-written article concerning the quality and traceability of extra virgin olive oil on the market.

The authors report fast, low-cost, effective and reproducible DNA extraction protocol, starting from previously reported protocols, including some modifications. The suitability and effectiveness of the proposed protocol were assessed by applying PCR-based fingerprinting methods, SSR and SNP genotyping. The developed method is proposed to be suitable for industrial/legal applications.

Figure 2 is missing in the manuscript.

Reviewer 3 Report

This is generally a well-written article on the authentication of olive oils by using the DNA-based approaches. Some comments for the authors to improve the manuscript are listed below:

Texts in Figure 1 and 3 must be enlarged to make it legible. Quality of figures 3 and 4 must be improved. Figure 2 is missing in the manuscript. Introduction must be improved by including more recent references, trends, and research findings on the topic. In addition, the novelty of the study should be presented. Add limitations and disadvantages of the developed method in Discussion section. It is advised to add the following references: a) J. Agric. Food Chem. 2015, 63, 2284–2295. b) https://doi.org/10.1002/9781119135340.ch32 c) https://doi.org/10.1002/lite.201600048 d) https://doi.org/10.3390/proceedings2131061 e) https://doi.org/10.1016/j.tifs.2019.07.045 Smaller and related paragraphs must be combined to make a big paragraph. Grammatical correction and spell checks are required throughout the manuscript.
